# Design of a Structural Health Monitoring System and Performance Evaluation for a Jacket Offshore Platform in East China Sea

**Hailin Ye [1,*], Chuwei Jiang [1], Feng Zu [1] and Suzhen Li [2]**

1    Beijing Special Engineering Design and Research Institute, Beijing 100028, China
2    College of Civil Engineering, Tongji University, Shanghai 200092, China
*    Correspondence: yeharry@163.com

**Abstract:** Offshore platform plays an important role in ocean strategy, and the construction of structural health monitoring (SHM) system could significantly improve the safety of the platform. In this paper, complete SHM system architecture design for offshore platform is presented, including the sensor subsystem, data reading and transferring subsystem, data administration subsystem, and assessment subsystem. First, the sensor subsystem is determined to include the structure information, component information, and vibration information monitoring of the offshore platform. Based on the monitoring target, three sensor types including incline sensor, acceleration sensor, and strain sensor are initially selected. Second, the assessment subsystem is determined to include safety monitoring and early warning evaluation using static measurements, overall performance evaluation based on frequency variation, and damage identification based on strain modal using strain monitoring. Overall performance evaluation based on frequency variation and damage identification based on Strain modal are illustrated. Finally, an offshore platform in the East China Sea is selected to establish a finite-element model to discuss the application and feasibility of the SHM system, the frequency variation due to scouring, corrosion, the growth of marine organisms, and temperature variation was investigated, and the overall performance of the platform was also evaluated. This work can provide a reference for installation and implementation of SHM system for offshore platform.

**Keywords:** offshore platform; structural health monitoring; finite-element analysis; performance evaluation

## 1. Introduction

Offshore platforms are a form of marine infrastructure; they are an important tool for understanding the ocean and for developing marine resources, and they are of great significance in strategic marine development. So far, the fixed steel structure jacket platform has become the most widely used offshore platform [1]. For example, many offshore platform jackets have been built in China's Bohai Bay, China's offshore exploration and development technology has reached the world-class level, and the continuous production increase of Bohai oil field has provided energy demand for China's rapid economic development. However, offshore structures are faced with complex marine environment, e.g., wind, waves, currents, ice, or other loads. In order to ensure the functional performance during the safe design service life, structural health monitoring (SHM) of offshore structures plays a major role in ensuring their safe operation [2].

Typical SHM methods are typically based on natural frequency or modal shape, frequency response function and modal strain energy. In recent years, the parametric methods, especially the modal strain energy -based methods, have often been applied for damage detection of marine structures [3]. Those SHM method are mainly based on the collected structural dynamic response data and feedback the changes of structural dynamic characteristics to determine the loss location and degree of offshore platform structure. However,

due to the disturbance of marine environment, it is not easy to obtain accurate results [4]. Therefore, the improved SHM methods rely heavily on the accuracy of the modal analysis, some data processing techniques have been proposed to overcome the problems that arise in the identification of modal parameters, e.g., mode correction, sensitivity to noise. Liu et al. [5] introduced the factors of mode correction and defined a new damage indicator based on the concept of modal strain energy to localize the damage of offshore wind turbines based on spatial incompleteness with better accuracy. Li et al. [6] introduced frequency information into the traditional Stubbs index method. The effectiveness of the improved method for damage localization was proved by the numerical example and laboratory study of jacket platforms. Chen et al. [7] presented a two-step approach to capture the dynamic response of an offshore jacket platform based on the real-time kinematic global navigation satellite system (RTKGNSS), which combined a Chebyshev filter and complementary ensemble empirical mode decomposition with adaptive noise (hereinafter referred to as CF-CEEMDAN). Haeri et al. [8] proposed a new approach for SHM of offshore jacket platforms, which used the measured ambient vibration responses and the corresponding readable natural frequencies and mode shapes of the structural system. Liu et al. [9] proposed a novel method to detect structural damages of offshore platforms based on grouping modal strain energy, which method divided the unit modal strain energy into axial tension-compression and bending. Liu et al. [10] proposed an iterative noise extraction and elimination method to solve the difficulty of modal parameter identification caused by contaminated high-energy components in measured signals. Mojtahedi et al. [11] proposed an improved model reduction-modal-based method for model updating and health monitoring of an offshore jacket-type platform. Tang et al. [12] presented a sampling rate selection of sensors method in offshore platform SHM based on vibration.

However, the complexity of the marine environment leads to the above traditional and improved SHM methods to still present challenges, e.g., incomplete and inaccurate modal information, noisy measurements. With the in-depth study of SHM for offshore platform, mathematical statistical methods have also been introduced, which is capable of damage diagnosis with limited, dispersed modal data. Wang et al. [13] validated a recently developed cross model cross mode (CMCM) model updating the method for an offshore platform structure by using experimental data. Hosseinlou and Mojtahedi [14] presented an algorithm method based on the CMCM in combination with an iterative procedure along with introducing a simplified method that is named Pseudo simplified (PS) model technique. Hosseinlou et al. [15] represented a methodology using the CMCM method in combination with an iterative procedure which uses limited, spatially incomplete modal information. Fathi et al. [16] proposed a new Bayesian model updating framework using incomplete frequency response function (FRF) data. In this methodology, the amount of data in the objective function is increased using FRF at different excitation frequencies to reduce the detrimental impacts of uncertainties on model updating and damage detection. Li and Huang [17] proposed a combined method of cross correlation and principal component analysis (PCA) to detect the damage under the influence of wave excitations. Yang et al. [18] proposed a combination method of GNSS and accelerometer to monitor the vibration of platform structures, which could complement each other's advantages and compensate for their shortcomings in the dynamic deformation monitoring of a structure. In the past few years, intelligent algorithm also has been used in numerous vibration-based structural damage detection approaches, e.g., genetic algorithm, artificial neural network (ANN), convolutional neural network (CNN), fuzzy logic system (FLS), long and short-term memory (LSTM), et al., which usually lead to considerable computational complexity problem for the damage detection in offshore platform SHM operations [19]. Mangal et al. [20] presented that an automated vibration monitoring method of detecting the occurrence and location of damage in offshore jacket platforms used artificial neural networks. Mojtahedi et al. [11] used FLS and probabilistic analysis in SHM for offshore jacket platform. Wang et al. [21] used substructural identification and genetic algorithms to

damage address on offshore jacket platforms. Zhou et al. [22] proposed a health monitoring method for analyzing offshore wind power structures based on a genetic algorithm and an uncertain analytic hierarchy process (AHP). Both Yang et al. [23] and Zhang et al. [24] developed CNN-LSTM combination method for model frequency identification used in SHM, and obtained good results. Bao et al. [25] proposed an SHM method combining the random decrement technique (RDT) with long and short-term memory (LSTM) networks for offshore structures. Bahootoroody et al. [26] developed an advanced risk-based maintenance (RBM) methodology for prioritizing maintenance operations by addressing fluctuations that accompany event data.

There are already many monitoring systems. However, the current work either focuses on the marine environment and load monitoring, or focuses on the direct measurement of displacement, deformation, strain, and other structural responses. In the design of this monitoring system, more attention was paid to the selection and layout of sensors and the acquisition of monitoring data, and little was involved in the further professional application of data. That is, effective structural state assessment methods and safety early warning strategies are proposed based on the monitoring data, there is also a lack of a mature monitoring and evaluation system to fully grasp the time-varying performance of offshore platform structures. In this study, a complete structural health monitoring system is designed, and a jacket offshore platform is taken as the monitoring object for example. Methods for providing early warnings of safety hazards and evaluation of the overall performance of marine structures were proposed and the results are presented.

## 2. Design of a Structural Health Monitoring System for a Jacket Offshore Platform

### 2.1. Monitoring System Composition

The purpose of SHM for offshore platform is to monitor the static and dynamic performance of a structure by collecting, processing, and analyzing data comprehensively and in real time. Such monitoring systems generally contain the sensor subsystem, data reading and transferring subsystem, data administration subsystem, and assessment subsystem, as shown in Figure 1. In this system, the function of the sensor subsystem is to obtain physical information from the offshore platform and convert it into electrical signals. The function of the data reading and transferring subsystem is to collect sensor signals and transfer them to the data administration subsystem. The data administration subsystem can collect, organize, and store the monitoring data, and the assessment subsystem evaluates and analyzes the collected data and gives feedback on structural safety information in time.

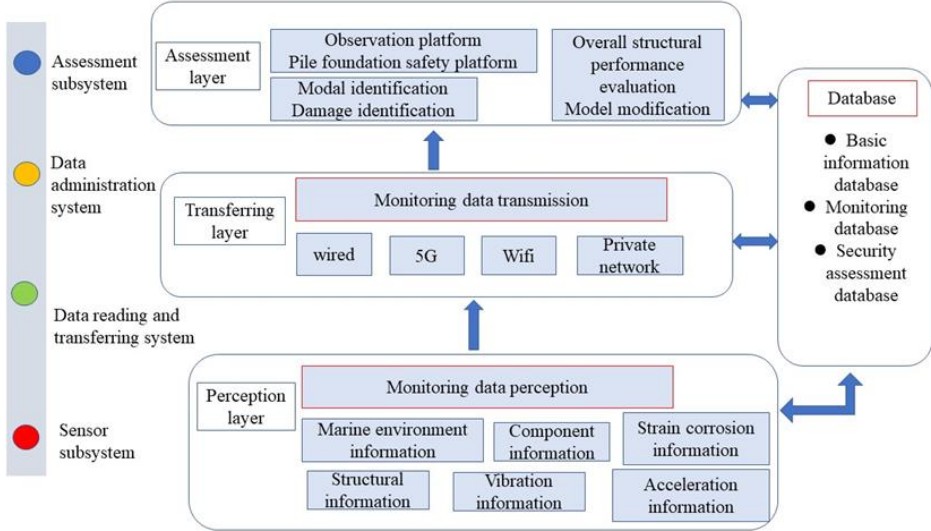

**Figure 1.** Schematic design of the offshore platform structural health monitoring system.

*2.2. Sensor Subsystem*

The first task in the construction of the sensor subsystem is to determine the selection and layout of the sensors. According to the structural characteristics of a jacket offshore platform, the types of sensors can be divided into those for: (a) offshore environmental information monitoring; (b) marine environmental information monitoring; (c) structure information monitoring; (d) platform component monitoring; and (e) vibration monitoring.

*2.3. Assessment Subsystem*

The assessment subsystem examines the overall performance of the offshore platform using real-time data, which includes safety monitoring and early warning evaluation using static measurements, overall performance evaluation based on frequency variation and damage identification based on strain modal, which are described below.

2.3.1. Safety Monitoring and Early Warning Evaluation Using Static Measurements

Safety monitoring and early warning evaluation of the offshore platform are based on static measurements. This evaluation mainly checks whether each index is within a defined safety threshold by measuring the platform displacement and inclination and the pile foundation load.

Under the action of various factors, an offshore platform will experience a certain degree of displacement and inclination. Within the safety threshold, this will not affect the service performance of the structure. However, once the displacements or loads exceed certain ranges, this will threaten the safety of the platform.

Platform Displacement Monitoring

Measurement of the displacement and its direction is realized by positions on the top of offshore platform using the GPS or Beidou system. As shown in Table 1, the risk to the offshore platform can be divided into three levels according to the UC value (the ratio of the maximum allowable stress under combined working conditions of member compression, tension, and bending). It is necessary to determine the relationship between the displacement of the observation platform and the UC range of its members. Many scholars have examined this according to structural collapse theory and historical data, and it will not be repeated here. For specific criteria, please refer to the relevant literature [27].

**Table 1.** Classification of structural failure modes.

| Failure Form | Consequence | Risk Level | Warning Color |
|---|---|---|---|
| First member UC exceeds 1.05 | Local strength reduction | Class A | Blue |
| Local member UC exceeds 1.05 | Local failure | Class B | Orange |
| First node invalid | Overall structural failure | Class C | Red |

UC: Maximum allowable stress ratio of member under combined working conditions of compression, tension, and bending.

Platform Tilt Monitoring

In the relevant technical specifications for jacket platforms, the platform inclination is generally required to be controlled within 0.5% [28]. The index and the early warning condition ($\beta_{\max}$) is calculated according to the following equation.

$$\beta_{\max} = |\theta_t - \theta_b|_{\max} < 0.5\% \tag{1}$$

where $\theta_t$ and $\theta_b$ are the inclination of the top and bottom of a guide leg as measured by its inclination sensors, respectively.

Pile Foundation Load Monitoring

The pile foundation load is mainly composed of the initial load and the variation of the load. The initial load and the dead weight of the steel piles can be regarded as fixed values, and the change in the load on the pile foundation can be obtained through monitoring [29].

Under normal sea conditions, the jacket structure can be regarded as a linear system, as shown in Figure 2. In this paper, the load change at the top of the jacket leg is denoted by $\Delta F$ and the load transfer function of the platform structure is $C$, the change in the pile foundation load $\Delta P$ can be expressed as

$$\Delta P = C \times \Delta F. \tag{2}$$

If the initial load is $P_0$ and the dead weight of the steel-pipe pile is $W$, the pile foundation load $P$ is

$$P = P_0 + \Delta P + W = P_0 + C \times \Delta F + W \tag{3}$$

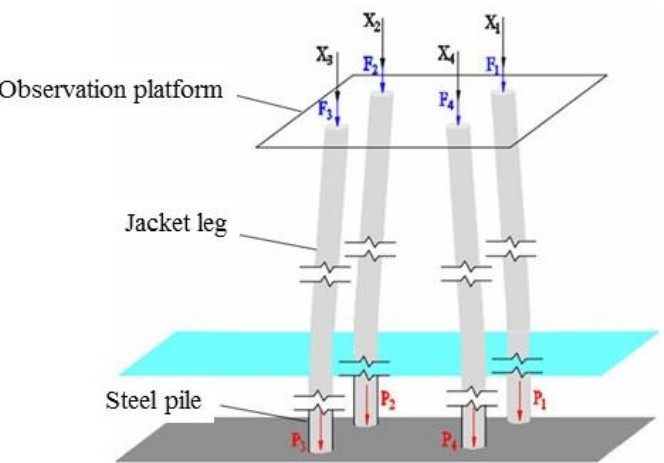

**Figure 2.** Schematic diagram of pile foundation load.

It is also necessary to determine the change in the load at the top of the conduit leg $\Delta F$ to obtain the change in the load at the pile foundation $\Delta P$. The stress model of conduit leg is shown in Figure 3, in which: $F$ is the load on the top of the conduit leg; $M_x$ and $M_y$ are the bending moments in the $x$ and $y$ directions, respectively, at the top of the leg; $\theta$ is the included angle between the conduit leg and the $x$–$y$ plane; and $\alpha$ and $\beta$ are the included angles between the diameter of any section position and the two coordinate axes, which can be obtained by the knowledge of material mechanics.

$$N = S \times E \cdot \frac{\varepsilon_1 + \varepsilon_2}{2} \tag{4}$$

where $N$ is the total pressure load, $E$ is the elastic modulus, $\varepsilon_1$ and $\varepsilon_2$ are the strain value at both ends of any diameter on the pile section.

The Equation (4) denotes that the load on the top of a guide leg can be obtained by measuring the strain of any two points on the diameter of its section; the change in the load on the top of the guide leg can then be obtained by changing Equation (4) to an incremental form:

$$\Delta F = -\frac{SE}{\sin \theta} \cdot \frac{\Delta(\varepsilon_1 + \varepsilon_2)}{2}. \tag{5}$$

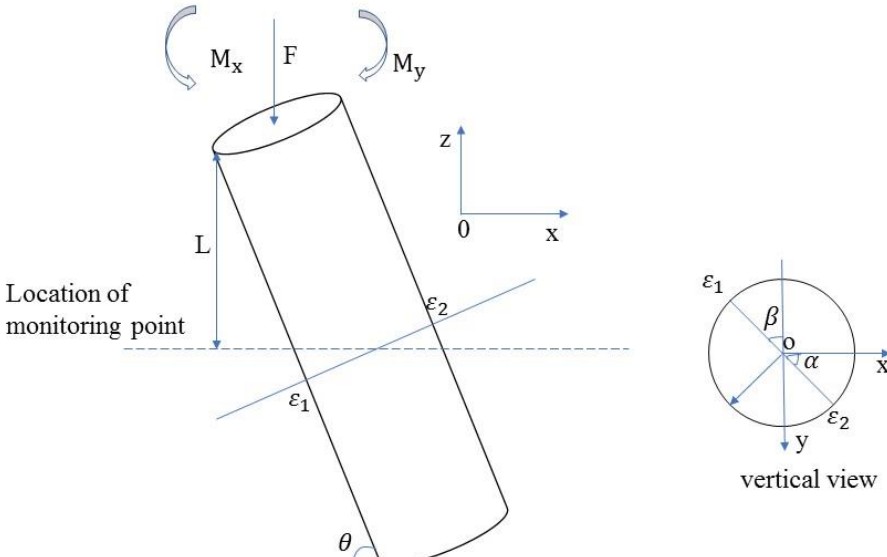

**Figure 3.** Stress model of a jacket leg.

Therefore, the following arrangement scheme can be adopted: two strain sensors are arranged in any diameter direction on the top section of each conduit leg. The change in the value of the load on the top of the conduit leg can be obtained by monitoring the change in the strain measured by the strain sensor, and this is then multiplied by the transfer function to obtain the change in the pile foundation load.

Finally, to prevent excessive pile load, it is necessary to determine a load range for safe operation of the pile foundation. In engineering, a safety factor is usually adopted to achieve this purpose. The monitoring and early warning conditions of the pile-end bearing capacity should be comprehensively evaluated according to different requirements. Here, a safety factor of 2.0 is used with reference to the American Petroleum Institute [30] specification; that is, the load on the pile foundation should be less than half of its allowable bearing capacity. This can be expressed as

$$P \leq [P] = \frac{P_m}{2} \tag{6}$$

where $P_m$ is the allowable load of pile foundation.

### 2.3.2. Overall Performance Evaluation Based on Frequency Variation

Under a variety of cyclic loads, the natural frequency of an offshore platform will change. Therefore, dynamic characteristics such as the natural frequency of the structure could be used to comprehensively checkout its overall performance. Modal identification is a process of extracting the characteristic natural vibration parameters of the structure—such as frequency, vibration mode, and damping—through real-time monitoring of response data. Overall performance evaluation based on frequency changes: it is an important index to evaluate the integrity and safety of the structure. The health of the structure can be investigated through changes in frequency.

Two modal parameters, frequency and strain modal shape, can be identified by strain sensor for damage identification in general. So, the natural frequency of the structure can be easily obtained through strain monitoring. Once the structural component is damaged, the health status of the structure can be evaluated by the natural frequency of the structure before and after the damage. The frequency change rate of the structure before and after damage can be defined as:

$$\frac{df_i}{f_i} = \frac{|f_{ui} - f_{di}|}{f_{ui}} \tag{7}$$

where $f_{ui}$ and $f_{di}$ respectively represent the *i*-th order natural frequency value of the structure after and before damage.

### 2.3.3. Damage Identification Based on Strain Modal

Damage identification is a very important link in the evaluation subsystem. It can identify whether local member damage occurs during the operation of offshore platform, and then locate and quantify the damage. Damage identification method based on strain modal has been employed and verified in previous research [31], and the strain modal can be obtained from strain monitoring, then, the damaged member can be located by determining the corresponding position of the mutation of the strain modal difference before and after the damage, and the magnitude of the amplitude can reflect the damage degree.

## 3. Application to a Case Study

SHM assessment based on actual monitoring data is a straightforward way to verify the effectiveness of the method for an already built offshore platform. However, for an offshore platform under construction, measured structural responses are not available yet, and the numerical analysis is therefore an alternative [11,32]. The Ansys 17.0 software was used for finite-element analysis to analyze the overall performance of the structure.

### 3.1. Monitoring Object

The platform of a three-dimensional comprehensive observation tower under construction in the East China Sea is selected as the case study, the design of the SHM system for this platform is described in detail. The water depth at the location of the jacket offshore platform is 70 m, and it is exposed to the water surface. Its total height is 114 m, and this includes three parts: the jacket 13.5 m, wind tower 100 m, and observation platform 0.5 m. The physical dimensions of jacket, observation platform and wind tower are shown in Figure 4, the structural element parameters of the offshore platform are shown in Table 2.

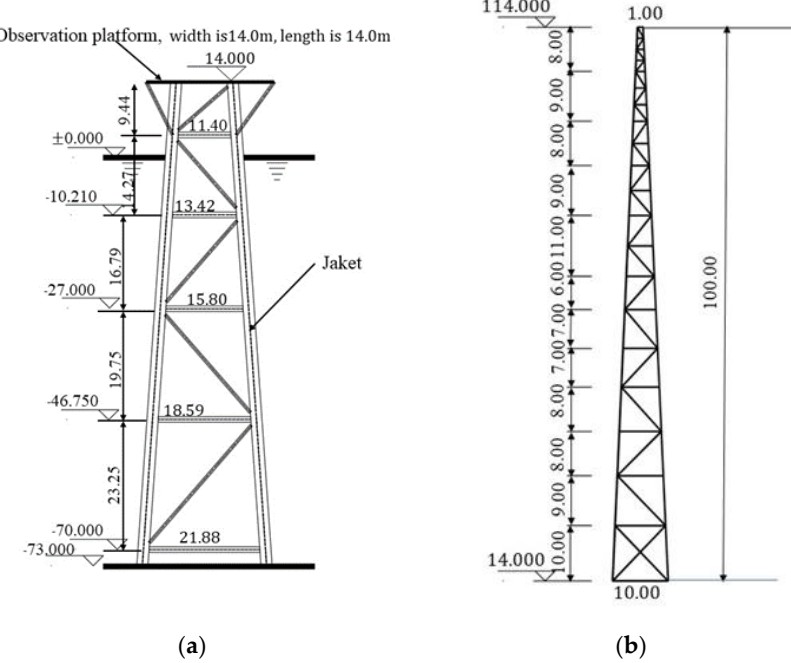

(a)    (b)

**Figure 4.** Physical dimensions of jacket offshore platform: (**a**) jacket and observation platform; (**b**) wind tower. (unit: m).

**Table 2.** Structural element parameters of offshore platform [33].

| Position | Description | Geometric Parameter | Material Parameters | | |
| --- | --- | --- | --- | --- | --- |
| | | | Elasticity Modulus | Density | Poisson' Ration |
| Jacket | Main Bar | Round steel pipe, diameter is 1.5 m, thickness is 0.06 m | 206 GPa | 7850 kg/m³ | 0.3 |
| | Interlayer transverse bracing | Round steel pipe, diameter is 0.8 m, thickness is 0.04 m | | | |
| | Diagonal bracing | Round steel pipe, diameter is 0.6 m, thickness is 0.025 m | | | |
| | Internal bracing | Round steel pipe, diameter is 0.6 m, thickness is 0.025 m | | | |
| Observation platform | beam | Rectangular steel tube, width is 0.4 m, height is 0.4 m, thickness is 0.02 m | | | |
| | deck | thickness is 0.02 m | | | |
| Wind tower | Structure leg | Round steel pipe, diameter is 0.5 m, thickness is 0.02 m | | | |
| | Transverse bracing | Round steel pipe, diameter is 0.25 m, thickness is 0.01 m | | | |
| | Diagonal bracing | Round steel pipe, diameter is 0.2 m, thickness is 0.0075 m | | | |

Based on the monitoring objectives, inclination sensors, strain sensors, linear polarization resistance probes, electrical resistance probes, and many other sensors are used. Because of limitations of space, only those sensors considered in the present work are shown in Figure 5. The principle of the layout the GPS stations is to set them at the center point of the platform and the open field of vision on the axis. The system is equipped with a total of eight inclination sensors (T1–T8): one at the top and one at the bottom of each of the four main pipes. The strain sensors (numbered S1-1, S1-2, S2-1, S2-2, S3-1, S3-2, S4-1, and S4-2) are divided into two types of location: the midpoints of each guide-leg rod layer, and at the top of each guide leg. Of these, four groups of strain sensors are set at the top of the four guide legs, and each group contains two strain sensors; their wires pass through the centers of the guide-leg sections.

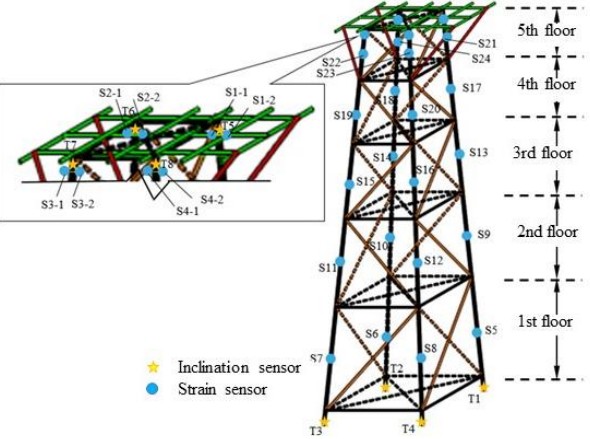

**Figure 5.** Sensor's placement on the jacket offshore platform.

*3.2. Establishment of Finite-Element Model*

For the general finite-element model by ANSYS, PIPE59 elements were used for the jacket above the mud surface, SHELL181 elements were used for the observation platform, BEAM4 elements were used for the platform beams and diagonal braces, and PIPE16 elements were used for the jacket below the mud surface. For numerical analysis, the model parameters must be defined properly [34,35], and the size of the model as well as the mesh size should be described [36,37], and the size of finite element mesh is 0.3 m with the 2-node linear rod element acceptable by ANSYS. The established finite-element model is shown in Figure 6.

In practice, the constraint conditions of the piles and soil will be changed by external factors, and this boundary is an important factor causing changes in the structural frequency. Therefore, it is of great significance to adopt appropriate boundary conditions. As shown in Figure 7, the soil reaction of a pile in the API specifications is divided into axial and lateral components to describe the establishment of the finite-element model under pile–soil interactions. The axial soil reaction includes the axial-transfer load reaction and the pile-tip load reaction. The horizontal spring constant is determined by the p–x curve, the vertical spring constant is determined by the t–z curve, and the end spring constant is determined by the Q–z curve [27].

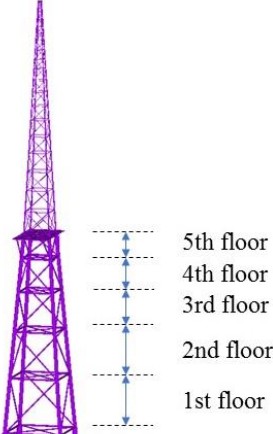

**Figure 6.** Finite-element model of the offshore platform.

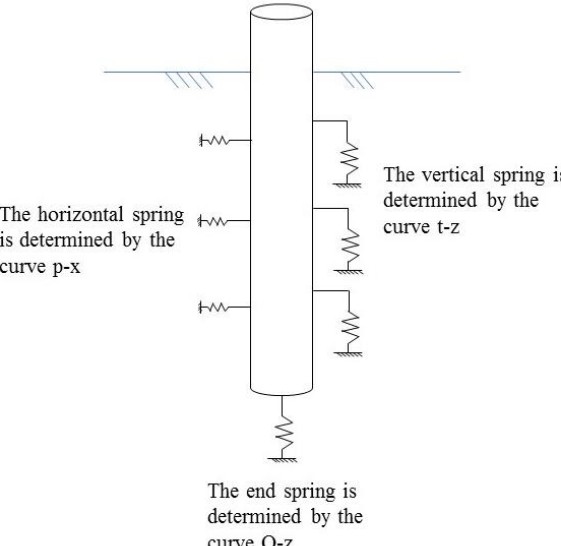

**Figure 7.** Pile–soil interaction model [30].

This paper only gives a brief description of the soft clay, for further details, please refer to the API specification. The p–x curve reflects the relationship between the soil force at the depth $z$ below the mud surface and the lateral deflection $x$ of the pile foundation at that depth under the action of a horizontal load. In soft clay, the p–x curve can generally be divided according to the load properties under static loads ($z < z_r$) and under cyclic loads ($z > z_r$).

Under a static load, the curve can be expressed as:

$$\frac{P}{P_u} = \begin{cases} 0.5\left(\frac{x}{x_c}\right)^{\frac{1}{3}} & (0 \leq x/x_c \leq 8) \\ 1 & (x/x_c \geq 8) \end{cases} \tag{8}$$

where $P_u$ is the ultimate soil resistance (unit: kPa); and $x_c$ can be determined according to $x_c = 2.5\varepsilon_c \times D$, in which $\varepsilon_c$ is the strain at 1/2 the maximum stress when an undrained compression test of an undisturbed soil sample is carried out in the laboratory and $D$ is the pile diameter (unit: mm). Under long-term cyclic loads, the curve can be determined according to Table 3.

**Table 3.** The p–x curve of soft clay under long-term cyclic loading.

| $z < z_r$ | | $z > z_r$ | |
|---|---|---|---|
| $P/P_u$ | $x/x_c$ | $P/P_u$ | $x/x_c$ |
| 0.00 | 0.0 | 0.00 | 0.0 |
| 0.50 | 1.0 | 0.50 | 1.0 |
| 0.72 | 3.0 | 0.72 | 3.0 |
| $0.72z/z_r$ | 15.0 | 0.72 | $\infty$ |
| $0.72z/z_r$ | $\infty$ | | |

The pile–soil interaction model used COMBIN39 spring elements, and PIPE16 elements were used for the steel-pipe piles. Since no actual information about the bottom layer is known, soil-layer information presented in a previous report was used for this analysis [38]. Finally, the finite element model of offshore platform with pile–soil interaction is established, and the first ten modal frequencies of the model are listed in Table 4.

**Table 4.** First ten frequencies of the model.

| Order | First | Second | Third | Fourth | Fifth |
|---|---|---|---|---|---|
| Frequency (Hz) | 0.524 | 0.525 | 0.918 | 1.053 | 1.054 |
| Order | sixth | seventh | eighth | ninth | tenth |
| Frequency (Hz) | 1.862 | 1.862 | 2.744 | 2.857 | 2.913 |

### 3.3. Overall Performance Evaluation Based on Frequency Variation

While a structural member is damaged, the mass matrix could be considered as unchanged, only changing the local stiffness, thus affecting the overall stiffness of the structure and affecting the natural frequency of the structure. The damage of structural members is simulated by changing the elastic modulus of structural members [11]. Based on the final element model and its natural frequency (Table 4) established in Section 3.2, the effect of damage on the overall health of the structure is discussed. The list of loss conditions established is shown in Table 5 below.

| Conditions | Stiffness Damage Degree | Location of Damaged Components |
|---|---|---|
| A (Single component damage) | 5%, 10%, 20%, 30%, 40%, 50% | Main pipe of Monitoring point s17 (4th floor) |
| B (Single component damage) | 5%, 10%, 20%, 30%, 40%, 50% | One of diagonal braces between 4th floor and 5th |
| C (Multi component damage) | 5%, 10%, 20%, 30%, 40%, 50% | Superposition of Condition A and Condition B |

The change of natural frequency change rate of each order of complex marine environment structure with damage degree is shown in Figures 8 and 9. It can be seen from Figure 8 that in the case of damage to a single member, the natural frequencies of each order of the platform are gradually increasing with the increase of damage degree. Condition A is more sensitive to the low order natural frequencies when the main member is damaged, and Condition B is more sensitive to the high order natural frequencies when the diagonal brace is damaged. At the same time, it can be seen that when the member is damaged to 50%, the effect on the natural frequencies of the offshore platform structure is only 1.3%. So, single adoption SHM based on frequency variation for offshore platform requires more ways than one because of complex marine environment, e.g., strain modal.

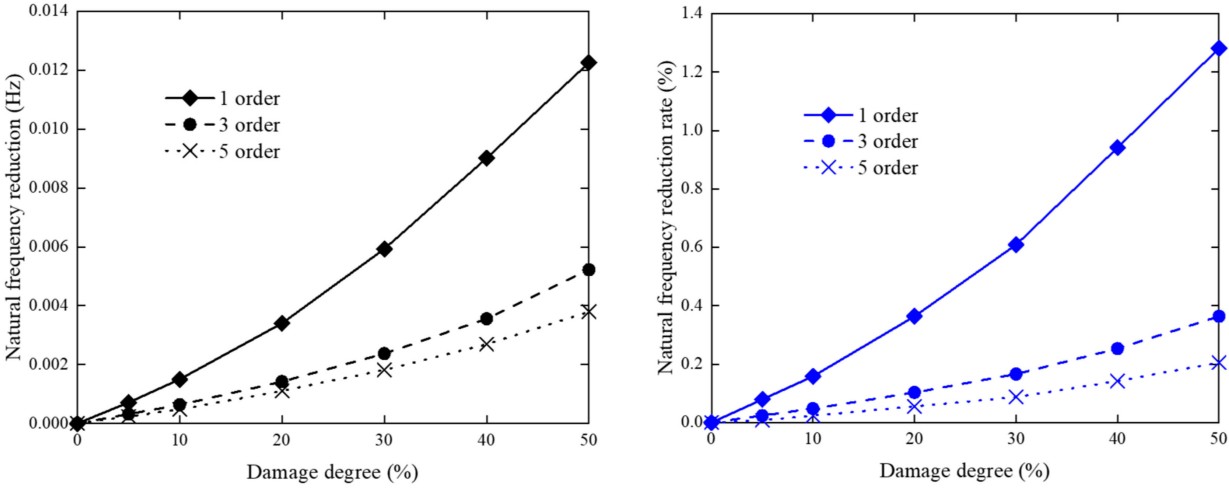

**Figure 8.** Structural natural frequency change under Condition A.

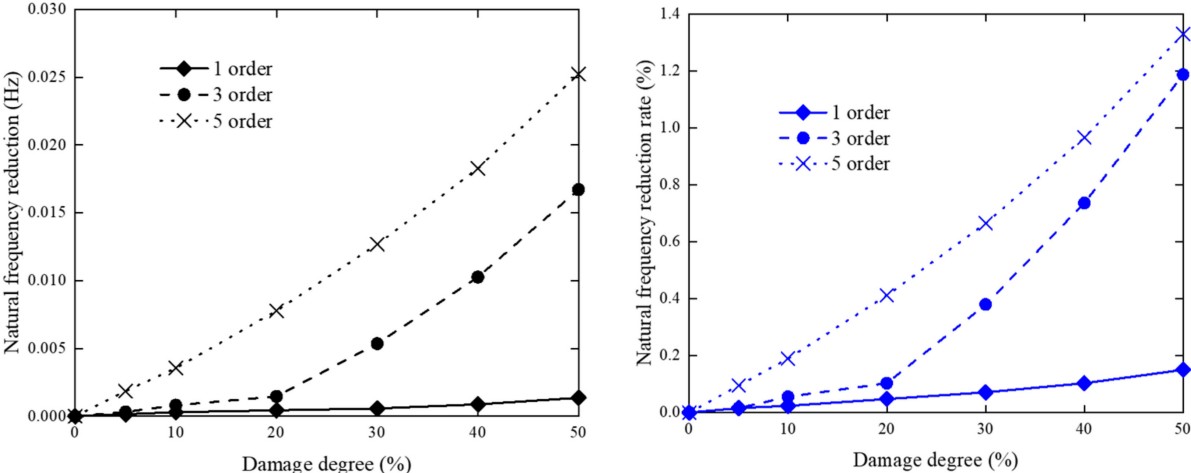

**Figure 9.** Structural natural frequency change under Condition B.

*3.4. Damage Identification of the Case Based on Strain Modal*

Based on the finite element model, analyzing the position corresponding to the mutation of the strain modal difference before and after the case damage in Table 5 to locate the damaged bar, and the magnitude of the amplitude can reflect the degree of damage. The first-order strain modal shapes of all bars before and after damage are plotted. Because of limitations of space, only the first-order strain difference diagram of 50% damage of the member were shown in Figure 10. We can see that first, compared with other layers of members, all members in the damage layer have more prominent and obvious changes in strain modal. Second, in the damage layer, the strain modal of the damaged member changes more sharply than that of other members, so the strain modal can not only locate the damage layer, but also accurately locate the damaged member.

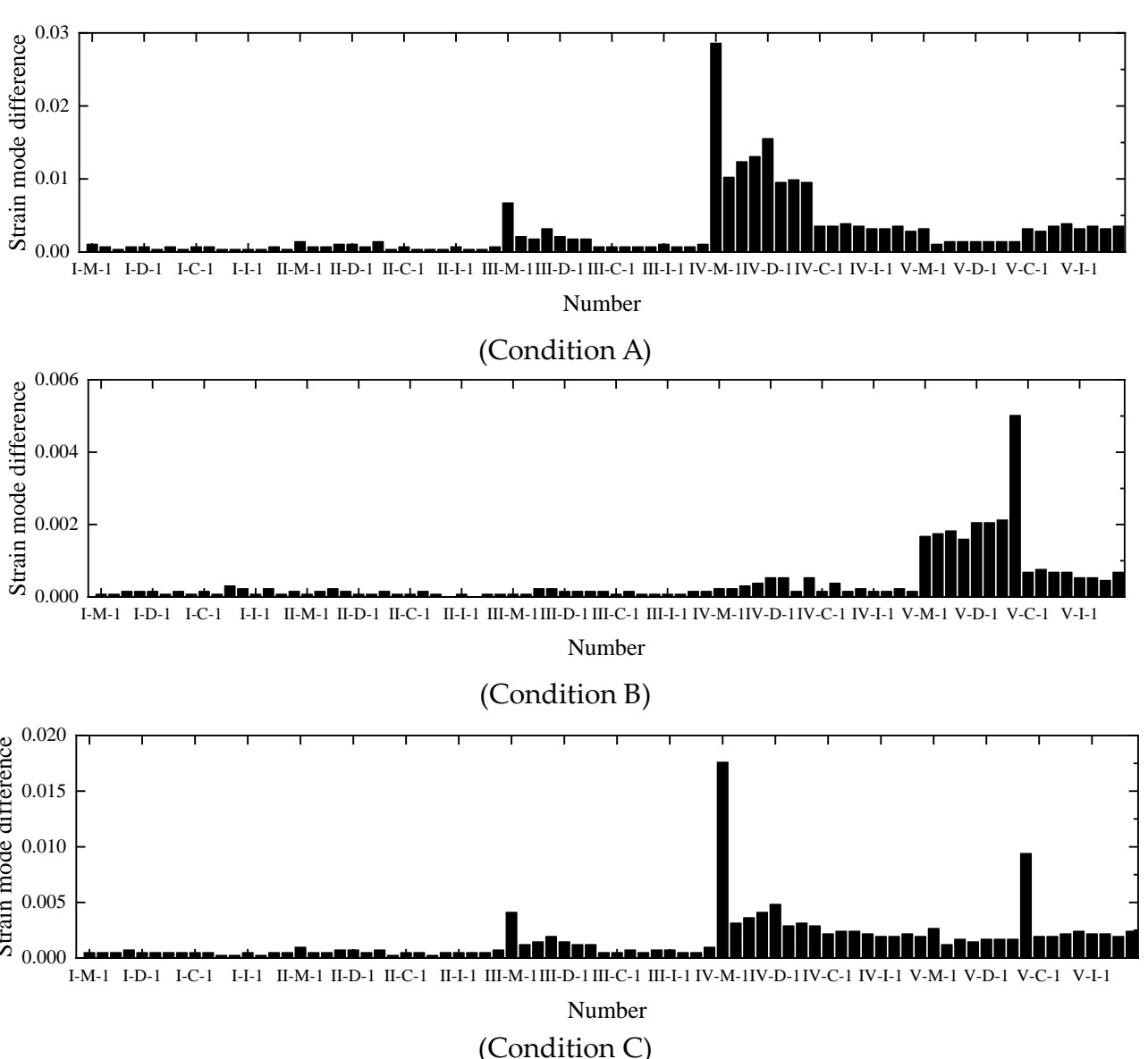

**Figure 10.** First order strain difference diagram of 50% bar damage.

*3.5. Analysis of the Factors Affecting the Health of an Offshore Platform Structure*

The natural frequency is a property of the structure that does not generally change with changes in external load. However, factors such as erosion and corrosion will cause the mass matrix and stiffness matrix to change, and this indirectly changes the natural

frequency of the structure. This section discusses the impacts of scouring, corrosion, marine organisms, and temperature.

Based on the finite element model established in Section 3.2, the effects of four actions on the model in six different years were considered. These four actions were scouring, corrosion, marine biological growth, and temperature, and the six time periods were 0, 5, 10, 15, 20, and 25 years. This is a total of 24 working conditions. For convenience of introduction, the three actions of scouring, corrosion, and marine biological growth are collectively referred to as external factors, and the effect of temperature is referred to as a climate factor. We extracted the natural frequencies of the model under the 24 working conditions to draw the diagram, and the time-varying effect on the frequency is shown in Figures 11–14.

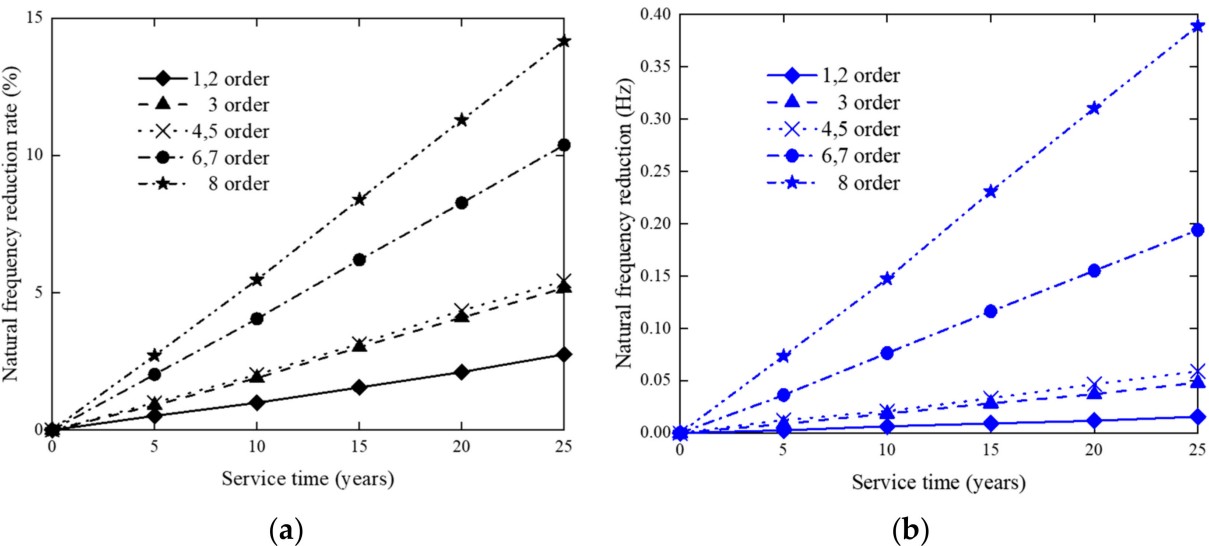

**Figure 11.** The diagram of time-varying effect on frequency under scouring. (**a**) Frequency change rate under scouring. (**b**) Frequency variation under scouring.

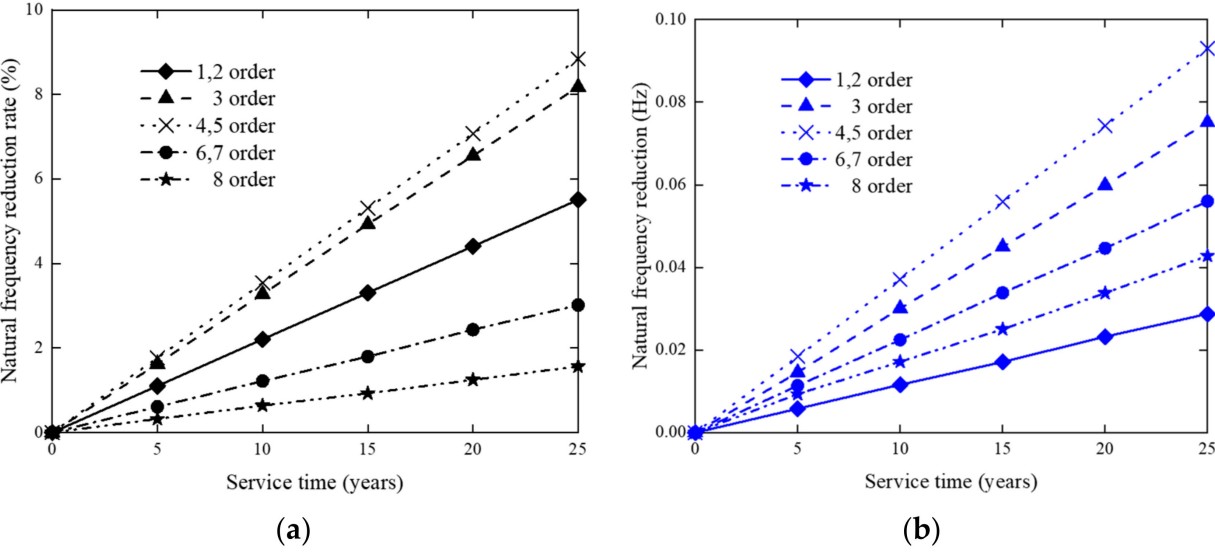

**Figure 12.** The diagram of time-varying effect on frequency under corrosion. (**a**) Frequency change rate under corrosion. (**b**) Frequency variation under corrosion.

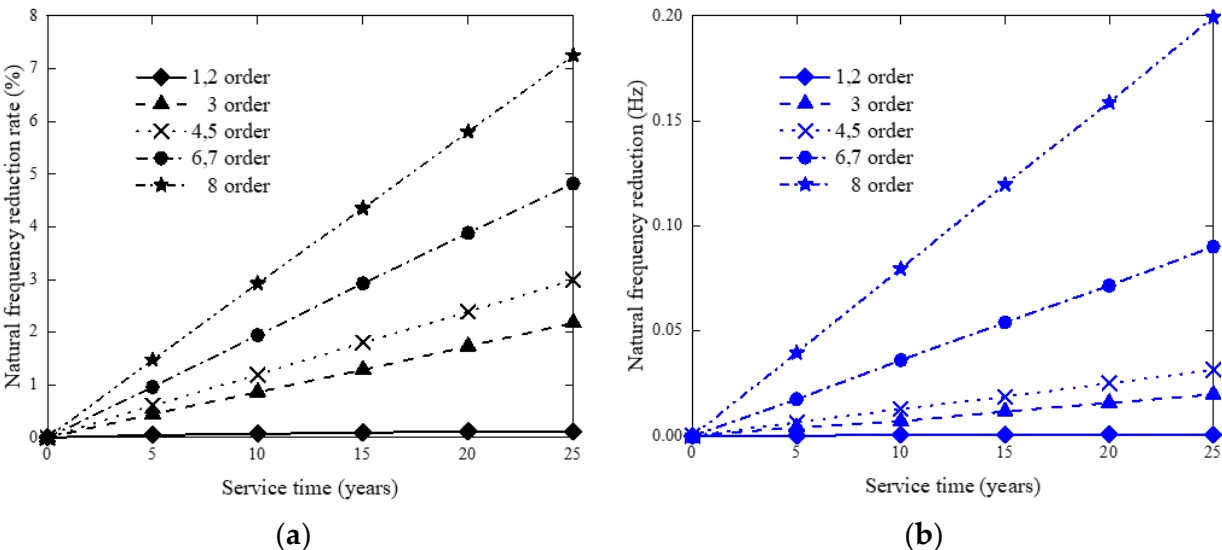

**Figure 13.** The diagram of time-varying effect on frequency under marine life growth. (**a**) Frequency change rate under marine life growth. (**b**) Frequency variation under marine life growth.

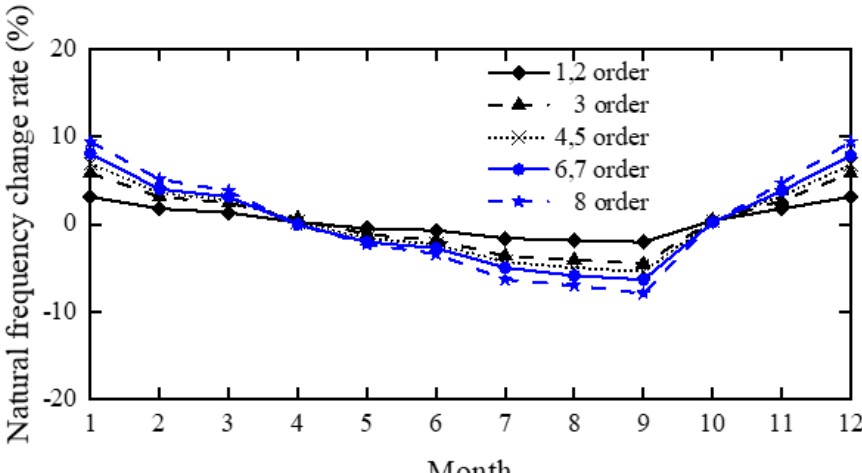

**Figure 14.** The diagram of overall natural frequency variation of the offshore platform over a whole year.

### 3.5.1. Scouring

Scouring is a natural phenomenon causing denudation. Local scouring causes the soil near the pile foundation to peel off; this reduces the penetration depth of the pile, reduces the constraint of the soil on the pile foundation, and reduces the lateral stiffness of the platform structure and thus affects its natural frequency. The change in the scouring depth over time can be expressed as [39]:

$$h(t) = h_i(t - t_0), \tag{9}$$

where $h(t)$ is the scouring depth at time $t$, $h_i$ is the annual scouring rate, and $t_0$ is the time at which scouring began.

As noted, scouring is the reduction of the soil depth at the pile foundations year by year; that is, a certain length of soil spring is deleted with increasing service time. Here, we assume that the depth of the steel pipe pile under the mud surface is reduced by 0.2 m every year [39].

The diagram of time-varying effect on frequency under scouring is shown in Figure 11. It shows that the platform frequency generally decreases with the increasing of service time. Within the service life, the effects of scouring on the frequency is 3–15%, so scouring has a greater impact on the higher-order modes.

### 3.5.2. Corrosion

With increasing service life, the diameter the of steel structure will decrease and pitting corrosion will occur under the action of seawater. This will affect its structural stiffness and natural frequency. Generally speaking, defining an accurate corrosion model is very complex; as such, we use a simple linear model here to describe the corrosion [40]:

$$r(t) = r_i(t - t_0),$$
(10)

where $r(t)$ is the material corrosion thickness at time $t$, $r_i$ is the annual corrosion rate, and $t_0$ is the time at which corrosion began.

Corrosion is realized by reducing the component wall thickness with increasing service time. Considering that the corrosion rates of surfaces in air and under the sea are different, it is assumed that the wall thickness of the platform components under the sea surface is reduced by 0.2 mm every year, while that of components exposed to air is reduced by 0.05 mm every year [40].

The diagram of time-varying effect on frequency under corrosion is shown in Figure 12. It denotes that the platform frequency generally decreases with the increasing of service time. Within the service life, the effects of corrosion on the frequency are 1–10%, so corrosion also has a greater impact on the lower-order modes.

### 3.5.3. Marine Life Growth

Over time, an increasing number of marine organisms will attach themselves to the structure and grow. This increases the weight of the underwater structure and reduces its overall natural vibration frequency. A simplified linear mass-increase model can be described by the expression [41]:

$$m(t) = m_i(t - t_0),$$
(11)

where $m(t)$ is the mass of marine life at time $t$, $m_i$ is the annual growth rate of marine life on the structure, and $t_0$ is the time at which the growth of marine life began.

As noted, the growth of marine life is realized through added mass. It is assumed that the growth rate of marine life on underwater components is 0.4 cm per year and that the added mass density is 1400 kg/m$^3$ [41].

The diagram of time-varying effect on frequency under marine life growth is shown in Figure 13. It illustrates that the platform frequency generally decreases with the increasing of service time. Within the service life, the effects of marine organisms on the frequency are 0–8%, so the impact of marine organisms is mainly on the higher order modes.

### 3.5.4. Temperature

The influence of temperature on the structure has two aspects: first, the elastic modulus of the material is negatively correlated with temperature, as shown in Table 6; second, the statically indeterminate structure produces secondary stress.

**Table 6.** Values of metal elastic modulus (*E*) at different temperatures [42].

| $T$ (°C) | 0 | 5 | 10 |
|---|---|---|---|
| $E$ ($10^5$ MPa) | 2.07 | 2.065 | 2.06 |
| $T$ (°C) | 15 | 20 | 25 |
| $E$ ($10^5$ MPa) | 2.055 | 2.05 | 2.045 |

The literature shows that changes in the axial size of slender members caused by temperature make little contribution to changes in their structural frequency [43]. Therefore, in this study, the influence of temperature on the frequency of the structure was considered only from the perspective of changes in elastic modulus.

The effect of temperature is simulated by reducing the elastic modulus of steel. For the observation platform and wind tower, which are above the sea surface, the temperature at the sea surface is used. The temperature of the jacket part is taken as the average temperature of the layer number in which the rod is located. The temperatures of the layers are listed in Table 7, and the elastic modulus values of steel corresponding to these temperatures are listed in Table 6. Elastic modulus values not directly listed were obtained by linear interpolation.

**Table 7.** Temperature of jacket offshore platform (unit: °C).

| Month | | 3 | 7 | 9 | 12 |
|---|---|---|---|---|---|
| | 1st floor | 4 | 6 | 8 | 12 |
| | 2st floor | 4 | 10 | 13 | 10 |
| Temperature (°C) | 3st floor | 4 | 15 | 16 | 10 |
| | 4st floor | 4 | 20 | 18 | 10 |
| | 5st floor | 4 | 20 | 18 | 10 |

As shown in Figure 14, the diagram of overall natural frequency variation of the offshore platform over a whole year. Except for the effect of temperature, the platform frequency generally decreases with the increasing of service time. Scouring and marine organisms had a greater impact on the higher-order modes, and corrosion had a greater impact on the lower-order modes. The climate factor, temperature, had a great influence on the higher-order modes, which showed a strong periodic regularity. The natural vibration frequency of the platform increases in spring and winter (December–March) and decreases in summer and autumn (June–October).

Based on the above analysis, the main factors causing frequency changes in different months of a year (vertical) and the same month of different years (horizontal) were considered. From this longitudinal analysis, it can be seen from Figure 14 that the frequency in a given year is mainly affected by the temperature in each month, and the influence of external factors is negligible. Therefore, the main influencing factor in the longitudinal analysis was temperature.

For simplicity, we divided the months seasonally, taking March, July, September, and December as being representative of the four seasons and observed the change rates of the low-order (1st–3rd order) natural frequency components corresponding to these months. From this analysis, we can see that the change range of low-order frequency components is 0–5%, and the change range of high-order (4th–8th) frequency components is 0–8%.

From horizontal analysis, we found that the influence of temperature on frequency in a given month is negligible, and the frequency is mainly affected by the external factors. Therefore, the main influence in the horizontal analysis was found to be the external factors. The division of the frequency variation range was consistent with that in the longitudinal analysis; only three external influencing factors were included. The final analysis results are listed in Table 8.

**Table 8.** Main factors causing the change of frequency.

| Frequency Variation (%) | Different Months of the Same Year | | Same Month of Different Years | |
|:---:|:---:|:---:|:---:|:---:|
| | Lower Orders | Higher Orders | Lower Orders | Higher Orders |
| <1% | T | T | M | N |
| 1–3% | T | T | M | C, M |
| 3–5% | T | T | S | M |
| >5% | N | T | C | M |
| >8% | N | N | C | C |
| >10% | N | N | N | S |

Note: "T" represents temperature; "M" indicates marine life; "S" stands for scouring; "C" stands for corrosion; "N" indicates none. Lower orders are 1–3; higher orders are 4–8.

## 4. Discussion and Conclusions

### 4.1. Discussion

Structural health monitoring is a prerequisite and indispensable to safety monitoring and hazard early warning of offshore platform, and the SHM system needs to be designed before an offshore platform is put into operation. This paper designed a complete SHM system and sensor layout scheme for offshore platform, so as to provide a basis for subsequent SHM system installation and implementation. The design scheme and theoretical basis of the SHM system as well as the assessment subsystem was described in detail. The innovation of this study lies in the selection and layout scheme of sensors for an offshore platform under construction combined with the application strategy of monitoring data, which can provide reference for the subsequent installation and implementation of SHM system.

As to the assessment of SHM system, actual monitoring data could make sense, but before the SHM system is put into operation, 3-D numerical simulation is usually employed to investigate its characteristics and performance under variety scenarios [32,44]. So, in this paper, offshore platform in East China Sea was selected to establish the numerical model, the application and performance of the proposed SHM system were illustrated based on numerical simulation. The factors affecting the health of an offshore platform structure were also analyzed. The results show that the designed SHM system is reliable theoretically and technically. Currently, the SHM system has not yet been installed, but our study can guide the selection and construction of SHM system for the offshore platform. Taking the offshore platform in East China Sea as a case study, the application and performance of the SHM system have been illustrated, and it would be further verified after monitoring data are obtained.

### 4.2. Conclusions

In this study, a SHM system for a jacket offshore platform under construction was designed. The content and application of the whole SHM, the sensor subsystem, and the evaluation subsystem were analyzed. The results can provide reference for SHM of offshore platforms in terms of analysis methods, equipment selection and overall performance evaluation. Changes in the natural vibration frequency of the jacket structure under the influence of scouring, corrosion, growth of marine life, and varying temperature were analyzed. The main contributions of this paper can be summarized as follows.

- According to the overall framework of the sensor subsystem, data reading and transferring subsystem, data administration subsystem, and assessment subsystem, a SHM system for jacket offshore platform under construction was designed.
- A method for safety monitoring and early warnings for an offshore platform based on static tests of the platform displacement, inclination, and loads of pile was presented.
- Overall performance evaluation based on frequency variation and damage identification based on strain modal using strain monitoring of platform was presented.

- Taking an offshore platform in the East China Sea as an numerical case study, the application of the SHM system was discussed, which verifies its feasibility. At last, variation in the natural frequency of the model under the influence of scouring, corrosion, the growth of marine organisms, and temperature variations was quantitatively analyzed. The main factors causing frequency variations in different months of a given year and the same month in different years were obtained.

**Author Contributions:** H.Y.: writing—original draft, supervision, investigation, methodology. C.J.: data curation, formal analysis. F.Z.: methodology, resources. S.L.: methodology, validation. All authors have read and agreed to the published version of the manuscript.

**Funding:** This research was funded by general project of military logistics scientific research fund of China under project No. 2019J013.

**Acknowledgments:** Thanks for three anonymous reviewers for their detailed and constructive comments, which greatly improved the quality of the manuscript.

**Conflicts of Interest:** The authors declare no conflict of interest.

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
