# Peer review of "Design of a Structural Health Monitoring System and Performance Evaluation for a Jacket Offshore Platform in East China Sea"

_applsci, doi:10.3390/app122312021_

Round 1

Reviewer 1 Report

Dear authors:

It is a good study. It should be accepted after minor revision.

1 There are already many monitoring systems. Please clarify the innovation of the research in the Introduction.

2 Why do you choose such a Monitoring Object? It is a scientific study rather than a case study. Please clarify the reason in Section 2.1.

3 Please provide the parameters of the numerical model as in previous research and cite related research [1-2].

Ref.:

[1]Modeling microcapsule-enabled self-healing cementitious composite materials using discrete element method. International Journal of Damage Mechanics, 2017, 26(2): 340-357.

[2]A numerical chemo-micromechanical damage model of sulfate attack in cementitious materials. International journal of damage mechanics, 2022, 31(10):1613-1638.

4 Please provide the size of the numerical model, as well as the mesh size as in previous research [3-4].

Ref.:

[3] Parametric Study of Hip Fracture Risk Using QCT-Based Finite Element Analysis. CMC-COMPUTERS MATERIALS & CONTINUA 2022, 71 (1) ,1349-1369.

[4] A finite element study on femoral locking compression plate design using genetic optimization method. JOURNAL OF THE MECHANICAL BEHAVIOR OF BIOMEDICAL MATERIALS, 2022, 131, 105202.

5 Please provide the citation of equations in the manuscript which are not developed by the authors.

Reviewer 2 Report

In this manuscript the authors provide a reference for the establishment of structural health monitoring systems for offshore platforms. Although the topic of the manuscript is interesting and worthy of investigation, there are several issues that should be resolved by the authors. 1. Please revise the Abstract Section in order to eliminate common knowledge and highlight in a more clear way the contribution of the presented research work. 2. The Literature Review Section should be further elaborated in order to explicitly discuss what has been done in similar research works highlighting the limitations and challenges, and how the presented manuscript contributes to the field. 3. It is recommended to reduce the mathematical equations to the bare minimum required. 4. The authors improve numerical results presentation for system. 5. Please make sure that the quality of the figures is acceptable. Concretely 300dpi is the threshold resolution. 6. The completeness of the literature review should be further elaborated with the addition of more recent and relevant publications such as: Reliability estimation of reinforced slopes to prioritize maintenance actions Bahootoroody, F., Khalaj, S., Leoni, L., ...Di Bona, G., Forcina, A. International Journal of Environmental Research and Public Health, 2021, 18(2), pp. 1–12, 373

Reviewer 3 Report

Dear Authors,

I have gone through the manuscript titled "Design of a Structural Health Monitoring System and a Performance Evaluation Method for a Jacket Offshore Platform". I found the paper interesting and it deserves publication. The paper relates to use of sensor based technology in monitoring structural health of offshore platforms. Its a topic, which is very important from Health and Safety Stand point. Its also a topic of importance to various oil and gas companies who spend a lot of money on offshore structural integrity.  

The paper describes methods based on sensor data to monitor static and dynamic movements of the tall offshore structures. Authors also state that their method can model corrosion and growth of marine life. I have some concerns related to the model of corrosion and growth of marine life. The model is, first, very simple and based on some assumptions, which are not backed up in the text. I would like authors to provide evidence or some reference to back up that corrosion under sea is 0.2mm in thickness every year and on surface is 0.05mm per year. Where are these numbers coming from ? experimental data ? Similarly some assumptions are made about marine life growth. Again what is the basis of these assumptions ? Authors should provide either reference or experimental evidence.  

Corrosion and Marine life growth is regional and can differ from one location to another, one offshore environment to another. Also the numbers quoted are specific to offshore China. May be paper title should be such that it can be clear that method presented is more applicable to offshore China conditions. 

Apart from this I did find at few places some spelling mistakes and misplacement of full stops and commas. It should be corrected.

With these changes I will be happy to recommend paper for publication.

Thanks
